# SHAKEDROP REGULARIZATION

**Yoshihiro Yamada, Masakazu Iwamura and Koichi Kise**
Graduate School of Engineering Osaka Prefecture University
1-1 Gakuen-cho, Naka-ku Sakai-shi, Osaka
`yamada@m.cs.osakafu-u.ac.jp,{masa,kise}@cs.osakafu-u.ac.jp`

## ABSTRACT

This paper proposes a powerful regularization method named *ShakeDrop regularization*. ShakeDrop is inspired by Shake-Shake regularization that decreases error rates by disturbing learning. While Shake-Shake can be applied to only ResNeXt which has multiple branches, ShakeDrop can be applied to not only ResNeXt but also ResNet, and PyramidNet in a memory efficient way. Important and interesting feature of ShakeDrop is that it strongly disturbs learning by multiplying even a negative factor to the output of a convolutional layer in the forward training pass. ShakeDrop outperformed state-of-the-arts on CIFAR-10/100. The full version of the paper including other experiments is available at `https://arxiv.org/abs/1802.02375`.

## 1 INTRODUCTION

Recent advances in generic object recognition have been brought by deep neural networks. After ResNet (He et al., 2016) opened the door to very deep CNNs of over a hundred layers by introducing the residual block, its improvements such as PyramdNet (Han et al., 2017a;b) and ResNeXt (Xie et al., 2017) have broken the records of lowest error rates.

On the other hand, in learning, they often suffer from problems such as vanishing gradients. Hence, regularization methods help to learn and boost the performance of such base network architectures. Stochastic Depth (ResDrop) (Huang et al., 2016) and Shake-Shake (Gastaldi, 2017) are known to be effective regularization methods for ResNet and its improvements. Among them, Shake-Shake applied to ResNeXt is the one achieving the lowest error rates on CIFAR-10/100 datasets (Gastaldi, 2017).

Shake-Shake, however, has following two drawbacks. (1) Shake-Shake can be applied to only multi-branch architectures (i.e., ResNeXt). (2) Shake-Shake is not memory efficient. Both drawbacks come from the same root. That is, Shake-Shake requires two branches of residual blocks to apply. If it is true, it is not difficult to conceive its solution: a similar disturbance to Shake-Shake on a single residual block. It is, however, not trivial to realize it.

The current paper addresses the problem of realizing a similar disturbance to Shake-Shake on a single residual block, and proposes a powerful regularization method, named *ShakeDrop regularization*. While the proposed ShakeDrop is inspired by Shake-Shake, the mechanism of disturbing learning is completely different. ShakeDrop disturbs learning more strongly by multiplying even a negative factor to the output of a convolutional layer in the forward training pass. In addition, a different factor from the forward pass is multiplied in the backward training pass. As a byproduct, however, learning process gets unstable. Our solution to this problem is to stabilize the learning process by employing ResDrop in a different usage from the usual. Experiments show ShakeDrop outperformed state-of-the-arts on CIFAR-10/100.

## 2 EXISTING REGULARIZATION METHODS

**Stochastic Depth** (Huang et al., 2016) is a regularization method for ResNet (He et al., 2016) given as $G(x) = x + F(x)$, where $x$ and $G(x)$ are the input and output of the residual block, respectively, and $F(x)$ is the output of the residual branch on the residual block. While ResNet opened the door to very deep CNNs of over a hundred layers by introducing the residual block, He et al. (2016) pointed

out that high error rates obtained by a 1202-layer ResNet on the CIFAR datasets (Krizhevsky, 2009) is caused by vanishing gradients. Stochastic Depth which overcame problems of ResNet makes the network apparently shallow in learning by dropping residual blocks stochastically selected. On the $l^{th}$ residual block from the input layer, the Stochastic Depth process is given as

$$G(x) = \begin{cases} x, & \text{if } b_l = 0 \\ x + F(x), & \text{otherwise (i.e., if } b_l = 1). \end{cases} \tag{1}$$

where $b_l \in \{0, 1\}$ is a Bernoulli random variable with the probability of $p_l$. Huang et al. (2016) shows the linear decay rule which defines $p_l$ as $p_l = 1 - \frac{l}{L}(1 - p_L)$, where $L$ is the number of all layers and $p_L$ is the initial parameter, worked well. Stochastic Depth can be introduced not only in ResNet but also in its improvements which have a single branch of residual blocks such as PyramdNet (Han et al., 2017a;b).

**Shake-Shake** (Gastaldi, 2017) is a powerful regularization method for improving ResNeXt, an improvement of ResNet, architectures. The basic architecture of ResNeXt is given as $G(x) = x + F_1(x) + F_2(x)$, where $F_1(x)$ and $F_2(x)$ are the outputs of the residual branches. The number of residual branches is not limited to 2, and the number is the most important factor to control the result. Shake-Shake decreased error rates than ResNeXt by random weighted average as following

$$G(x) = \begin{cases} x + \alpha F_1(x) + (1 - \alpha)F_2(x), & \text{on forward pass} \\ x + \beta F_1(x) + (1 - \beta)F_2(x), & \text{on backward pass}, \end{cases} \tag{2}$$

where $\alpha$ is a random coefficient given as $\alpha \in [0, 1]$ and another random coefficient $\beta$ given as $\beta \in [0, 1]$. On the test time, the $0.5$ is used instead of the coefficient $\alpha$.

## 3 PROPOSED METHOD

In the forward pass, Shake-Shake interpolates the outputs of two residual branches (i.e., $F_1(x)$ and $F_2(x)$) with a random variable $\alpha$ that controls the degree of interpolation. As DeVries & Taylor (2017a) demonstrated that interpolation of two data in the feature space can synthesize reasonable augmented data, the forward pass of Shake-Shake can be regarded as doing something similar. Use of a random variable $\alpha$ generates many different augmented data. In order to realize a similar regularization to Shake-Shake on 1-branch network architectures, in the forward pass, we need a mechanism, different from interpolation, to synthesize augmented data in the feature space. Actually, DeVries & Taylor (2017a) demonstrated not only interpolation but also noise addition in the feature space works well.

Hence, following Shake-Shake, we apply random perturbation, using $\alpha$ and $\beta$, to the single output of a residual branch. In the backward pass, we can use the same way as Shake-Shake even on 1-branch network architectures. We call the regularization method mentioned above *1-branch Shake*. While it is expected to realize powerful generalization like Shake-Shake, by applying it to 110-layer PyramidNet with $\alpha \in [0, 1]$ and $\beta \in [0, 1]$ following Shake-Shake, the result on the CIFAR-100 dataset was hopelessly bad (i.e., an error rate of 77.99%). The failure is caused by too strong perturbation. However, weakening the perturbation would also weaken the effect of regularization. Thus, we need a trick to promote learning under strong perturbation.

Our idea is to use the mechanism of ResDrop for solving the issue. In our situation, however, the original usage of ResDrop (ResDrop promotes learning by making a network apparently shallow) does not contribute because a shallower network to which *1-branch Shake* is applied would also suffer from strong perturbation. Thus, we use the mechanism of ResDrop as a probabilistic switch of two network architectures: the original network (e.g., PyramidNet) and the one to which *1-branch Shake* is applied (e.g., PyramidNet + *1-branch Shake*).

Finally, we propose a new regularization method named *ShakeDrop*, which is given as

$$G(x) = \begin{cases} x + \alpha F(x), & \text{if } (b_l = 0 \ \wedge \ \text{on forward pass}) \\ x + \beta F(x), & \text{if } (b_l = 0 \ \wedge \ \text{on backward pass}) \\ x + F(x), & \text{otherwise (i.e., if } b_l = 1), \end{cases} \tag{3}$$

where $\alpha$ and $\beta$ are mutually independent random coefficients. On the test time, the $p_l$ is used instead of the coefficient $\alpha$.

Table 1: Top-1 errors (%) at the final epoch (300th or 1800th) on the CIFAR-10/100 datasets. Representative methods and the proposed ShakeDrop applied to PyramidNet are compared. "Reg" represents regularization methods including ResDrop (RD), Shake-Shake (SS) and proposed ShakeDrop (SD). If "Cos" is checked, 1800-epoch cosine annealing schedule (Loshchilov & Hutter, 2016) is used following Gastaldi (2017). Otherwise, 300-epoch multi-step learning rate decay schedule is used following each method. If "Fil" is checked, the data augmentation used in Cutout (CO) (De-Vries & Taylor, 2017b) or Random Erasing (RE) (Zhong et al., 2017), which randomly fills a part of learning images, is used. $*$ indicates the result is quoted from the literature. $+$ indicates the result is quoted from Gastaldi (2017). Compared to the same condition of Cutout, the state-of-the-art, the proposed method reduced the error rate by 0.25% on CIFAR-10 and 3.01% on CIFAR-100.

| Method | Reg | Cos | Fil | Depth | #Param | CIFAR -10 (%) | CIFAR -100 (%) |
|---|---|---|---|---|---|---|---|
| Coupled Ensemble (Dutt et al., 2017) | | | | 118 | 25.7M | *2.99 | *16.18 |
| | | | | 106 | 25.1M | *2.99 | *15.68 |
| | | | | 76 | 24.6M | *2.92 | *15.76 |
| | | | | 64 | 24.9M | *3.13 | *15.95 |
| | | | | - | 50M | *2.72 | *15.13 |
| | | | | - | 75M | *2.68 | *15.04 |
| | | | | - | 100M | *2.73 | *15.05 |
| ResNeXt (Xie et al., 2017) | | ✓ | | 26 | 26.2M | +3.58 | - |
| | | | | 29 | 34.4M | - | +16.34 |
| ResNeXt + Shake-Shake (Gastaldi, 2017) | SS | ✓ | | 26 | 26.2M | *2.86 | - |
| | | | | 29 | 34.4M | - | *15.85 |
| ResNeXt + Shake-Shake + Cutout (DeVries & Taylor, 2017b) | SS | ✓ | CO | 26 | 26.2M | *2.56 | - |
| | | | | 29 | 34.4M | - | *15.20 |
| PyramidNet (Han et al., 2017b) | | | | 272 | 26.0M | *3.31 | *16.35 |
| | | ✓ | RE | 272 | 26.0M | 3.42 | 16.66 |
| PyramidDrop (Yamada et al., 2016) | RD | | | 272 | 26.0M | 3.83 | 15.94 |
| | RD | ✓ | RE | 272 | 26.0M | 2.91 | 15.48 |
| PyramdNet + ShakeDrop (Proposed) | SD | | | 272 | 26.0M | 3.41 | **14.90** |
| | SD | | RE | 272 | 26.0M | 2.89 | **13.85** |
| | SD | ✓ | | 272 | 26.0M | 2.67 | **13.99** |
| | SD | ✓ | RE | 272 | 26.0M | **2.31** | **12.19** |

## 4 EXPERIMENTS

The proposed ShakeDrop applied to PyramidNet was compared with state-of-the-arts on the CIFAR-10/100 datasets. State-of-the-art methods introduced some techniques that can be applied to many methods in the learning process. One is *longer learning*. While most of methods related to ResNet use 300-epoch scheduling for learning, Shake-Shake use 1800-epoch cosine annealing, on which the initial learning rate is annealed using a cosine function without restart (Gastaldi, 2017). Another one is *image preprocessing*. DeVries & Taylor (2017b) and Zhong et al. (2017) showed that accuracy is improved by data augmentation which randomly fills a part of learning images. For fair comparison with these methods, we also applied them to the proposed method. We used the best coefficients of the proposed method found in preliminary experiments, where $\alpha \in [-1, 1]$ and $\beta \in [0, 1]$. It was surprising that $\alpha \in [-1, 1]$ performed better than $\alpha \in [0, 1]$, while a negative $\alpha$ means the output of the residual function oppositely changes. Table 1 shows the error rates. The proposed method, "PyramidNet + ShakeDrop," without *longer learning* and *image preprocessing*, was 3.41% on the CIFAR-10 dataset and 14.90% on the CIFAR-100 dataset.

Implementation details and more experiments using various base network architectures on some datasets are available in the full version of the paper. On these experiments, ShakeDrop regularization improved error rates on almost all networks.

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
