# OpenReview forum: "ShakeDrop regularization"
_ICLR.cc/2018/Workshop — Accept_

### Official Review · AnonReviewer1 · 2018-03-11

**Rating:** 6
**Confidence:** 5

**Review:**

The paper proposed a regularization method that is an extention of the shake-shake regularization applied on ResNext. The original shake-shake method is applied on 2 residual paths. The proposed method combines shake-shake and pyramid drop, such that it could be used on a single residual path. The method achieved state-of-art results on cifar-10/ 100 datasets.

I would be interested to see some analysis and discussions on why the proposed methods works well. The writing of the paper could also be improved.

---

### Official Review · AnonReviewer3 · 2018-03-11
**good experimental results, presentation could be improved**

**Rating:** 6
**Confidence:** 2

**Review:**

A new type of regularization technique for deep residual networks is proposed.  This paper is inspired by techniques which disturb the training by applying multiplicative factors to the convolutional layer outputs such as Shake-Shake (Gastaldi '17) and PyramidDrop (Yamada '16).  In the proposed approach, it is randomly sampled to either follow the standard variant of Pyramid net, or apply a variant of shake-shake to pyramid net.

The paper shows excellent experimental results on CIFAR-10 and CIFAR-100, surpassing the standard techniques and regularizers as well as the Shake-shake and pyramid net baselines.
- Clarity: some statement are not always so clear and do not always follow logically.
- Quality: given the space limitations, it is of course very difficult to include sufficient experimentation to explain the effects of the regularizer, but one or two sentences explaining this would be helpful to the reader.

---

### Official Review · AnonReviewer2 · 2018-03-14
**Sufficiently novel idea with strong empirical results**

**Rating:** 7
**Confidence:** 3

**Review:**

Sufficiently novel idea with strong empirical results
The authors present a new method for regularization, ShakeDrop, that demonstrates improved classification on both CIFAR-10 and CIFAR-100.  The method takes inspiration from Shake-Shake and Stochastic Dropout to develop an approach that works well for single branch ResNets.  This approach multiplies the residual by a random weight coefficient to add multiplicative noise to the system.  Alone, this added term destabilizes training.  To address this issue, random weighting itself is randomly sampled otherwise the standard residual term is used.

There is an interesting interplay between the random sampling of the random weight coefficients. It appears to the reviewer, that the combination is a random variable that is biased to sample \alpha and \beta equal to one.  The author's should explore their formulation more thoroughly to see if there is some underlying phenomena at play.

The paper is motivated from Shake-Shake, but while Shake-Shake's intent is for data augmentation through interpolation, ShakeDrop appears to learn robust, scale invariant features.  This should be emphasized in the text.

Overall, the paper presents an sufficiently novel idea with strong empirical results, which supports publication at a workshop.

Comments:
The reviewer assumes b_l for ShakeDrop is derived as for Stochastic Dropout (as referenced at the beginning of the paper) and \alpha and \beta are sampled from a uniform distribution as from Shake-Shake. If this is not the case, please clarify in the paper.

You note that \alpha = [-1, 1] performing better than [0, 1] was surprising.  If possible, please deduce an explanation for the difference in expected versus observed performance.

---

### Public Comment · ~Oriol_Vinyals1 · 2018-02-17
**Please Fix Length**

Your paper violates by a few lines the 3 page limit (see https://iclr.cc/Conferences/2018/CallForWorkshops). Please send us a fixed version of your PDF at iclr2018.programchairs@gmail.com by the end of Monday, February 19th, or else we will reject your paper.

Thanks,
ICLR2018 Program Chairs

---

### Decision · Program_Chairs · 2018-03-20
**ICLR 2018 Workshop Acceptance Decision**

**Decision:**

Accept

**Comment:**

Congratulations, your paper was accepted to the ICLR workshop.